# The Optimization and Biological Significance of a 29-Host-Immune-mRNA Panel for the Diagnosis of Acute Infections and Sepsis

**DOI:** 10.3390/jpm11080735

**Published:** 2021-07-28

**Authors:** Yudong D. He, Eric M. Wohlford, Florian Uhle, Ljubomir Buturovic, Oliver Liesenfeld, Timothy E. Sweeney

**Affiliations:** Inflammatix, Inc., 863 Mitten Rd, Suite 104, Burlingame, CA 94010, USA; yhe@inflammatix.com (Y.D.H.); ewohlford@inflammatix.com (E.M.W.); fuhle@inflammatix.com (F.U.); lbuturovic@inflammatix.com (L.B.); oliesenfeld@inflammatix.com (O.L.)

**Keywords:** biomarkers, diagnosis, prognosis, acute infections, sepsis, bacterial, viral, host response, immune response, interferon, neutrophil

## Abstract

In response to the unmet need for timely accurate diagnosis and prognosis of acute infections and sepsis, host-immune-response-based tests are being developed to help clinicians make more informed decisions including prescribing antimicrobials, ordering additional diagnostics, and assigning level of care. One such test (InSep™, Inflammatix, Inc.) uses a 29-mRNA panel to determine the likelihood of bacterial infection, the separate likelihood of viral infection, and the risk of physiologic decompensation (severity of illness). The test, being implemented in a rapid point-of-care platform with a turnaround time of 30 min, enables accurate and rapid diagnostic use at the point of impact. In this report, we provide details on how the 29-biomarker signature was chosen and optimized, together with its molecular, immunological, and medical significance to better understand the pathophysiological relevance of altered gene expression in disease. We synthesize key results obtained from gene-level functional annotations, geneset-level enrichment analysis, pathway-level analysis, and gene-network-level upstream regulator analysis. Emerging findings are summarized as hallmarks on immune cell interaction, inflammatory mediators, cellular metabolism and homeostasis, immune receptors, intracellular signaling and antiviral response; and converging themes on neutrophil degranulation and activation involved in immune response, interferon, and other signaling pathways.

## 1. Introduction

Sepsis is a syndrome most recently redefined (‘Sepsis-3′) as a life-threatening dysregulated host immune response to infection [1]. A leading cause of morbidity and mortality, sepsis is also a major driver of health system costs across the United States [1,2,3,4]. Although frequently used synonymously with bacteremia, sepsis can be caused by bacterial as well as viral, fungal, or parasitic infections; and also can be mimicked by noninfectious etiologies, making diagnosis particularly challenging when an infectious source is not immediately apparent [5]. Early treatment with appropriate antibiotics has been shown to reduce morbidity and mortality in cases of bacterial sepsis [6,7]. However, antibiotics are well known to cause substantial adverse side effects and may often be unwarranted [8,9]. Also, indiscriminate and prolonged use of antibiotics can lead to antimicrobial resistance [10]. Taken together, timely accurate diagnosis of sepsis, and its underlying cause, is of great clinical interest [11,12].

An ideal diagnostic test for the management of patients with acute infections and suspected sepsis should answer the following clinically actionable questions in a highly accurate and rapid (<30 min) manner: (1) does the patient require antibiotics? (2) what other diagnostic tests should be ordered to make the diagnosis? (3) what level of care does the patient need? Importantly, such a test would help the emergency department (ED) to: (1) avoid antibiotics in patients without bacterial infections (i.e., ‘rule-out’ bacterial infection); (2) identify bacterial infections early in patients clinically suspected of viral or noninfectious inflammatory conditions (i.e., ‘rule-in’ bacterial infection); (3) avoid extensive diagnostic workup in patients without viral infections (i.e., ‘rule-out’ COVID-19 and other severe viral infections); (4) determine the optimal level of care (e.g., observation instead of admission or discharge instead of observation) for a patient with lower severity (de-escalate); and (5) predict organ dysfunction and decompensation over a short period to decide on closer monitoring and possibly intensive care unit (ICU) transfer (escalate).

The InSep™ test for acute infections and sepsis is designed explicitly to meet these needs. As previously described [13], InSep quantifies 29 host-mRNAs on a specialized platform from whole blood sample in less than 30 min and reports results on (1) likelihood of a bacterial infection, (2) likelihood of a viral infection, and (3) severity of the condition (risk of need for ICU-level care within seven days). The two core components of sepsis (an acute infection and a dysregulated host response leading to organ dysfunction) can thus be separately identified within the same test. A panel of clinical experts concluded that innovative diagnostic solutions such as the InSep test could improve management of patients with suspected acute infections and sepsis in the ED, thereby lessening the overall burden of these conditions on patients and the healthcare system [13].

InSep and its core algorithms, IMX-BVN (InflamMatiX Bacterial-Viral-Noninfected)-3, which produces the bacterial and viral scores and IMX-SEV (InflamMatiX SEVerity)-3, which produces the severity score, use machine learning to interpret the levels of 29 host immune mRNAs [14]. The 29 mRNAs were initially selected from three nonoverlapping mRNA-based scores, consisting of (1) 11 mRNAs diagnosing the presence or absence of an acute infection [15], (2) 7 mRNAs for distinguishing an infection between bacterial and viral [16], and (3) 12 mRNAs for determining the risk of 30-day mortality from sepsis [17], all using a well-established multi-cohort analysis approach [18]. The public and private databases mined for the purposes above included heterogeneous cohorts from diverse geographies, ethnicities, age groups, diagnoses, and settings of care, including outpatient, emergency department, inpatient wards, and intensive care units.

Previous research-based (non-clinical) versions of InSep utilized the NanoString nCounter™ platform for quantitative high-multiplex mRNA measurements. However, the nCounter platform returns results in about 24 h, making it unsuitable for use for rapid workflows. Inflammatix thus developed the Myrna™ platform for rapid cartridge-based isothermal quantitative reverse-transcribed loop-mediated amplification (qRT-LAMP) profiling of mRNA [13]. Our analytical assay studies showed that some mRNA targets in early versions of InSep [19] were not easily quantified using LAMP, and so the 29 markers for InSep had to be partially reselected to ensure adequate analytical performance on the rapid cartridge.

Here, we report the analytical and bioinformatics basis for the marker swap, the new InSep 29-mRNA set, and an analysis of the final 29 genes and their biological significance. Understanding the pathobiology underpinning the clinical accuracy of InSep is key to its generalizability to clinical use.

## 2. Methods

### 2.1. Gene Expressopm Quantification with qRT-LAMP Platform

The InSep system performs relative quantification of multiple mRNAs through a split-well spatial multiplex approach, where each of 64 wells measures a single target. The qRT-LAMP system has been shown to have a 5-orders of magnitude dynamic range with single-fold-change resolution and 50-fold improved limit of detection vs. off-the-shelf LAMP systems. While this assay system works for any target, the LAMP primer system is complex and requires optimization to limit off-target amplification and primer-dimers. Optimization is also required for efficiency, speed, and tight inter-sample correlation with gold standard techniques.

### 2.2. Marker Swap Rational, Procedures, and Candidates

In addition to our 29 genes initially selected for InSep (Appendix A), we first compiled a list of 64 genes from prior studies that were shown to be diagnostic or prognostic in sepsis (Appendix A) and then profiled them together across Inflammatix’s sample bank using a single NanoString nCounter SPRINT Profiler capture and reporter code set. For each of the 64 alternative biomarkers, the median, 5th, and 95th percentiles of abundance were calculated. Markers with fewer than 400 copies (minimum 100 copies × 4 fold-change) at 95th percentiles were first excluded to ensure that sufficient abundance could be detected by RT-LAMP in different cohorts. Next, markers with lower than 4-fold dynamic change between the 95th and 5th percentiles were further excluded to minimize the number of markers with limited resolution.

Using this two-step exclusion selection method, we evaluated the original 29 markers (Appendix A) and found that 23 markers had both 95th percentile > 400 copies and 95th/5th fold change > 4. From this two-step exclusion selection method, 27 additional markers were identified out of 64 alternative candidates (Appendix A). Of these 27 markers, 19 had five-fold or high dynamic change and were ranked as Tier 1, while the remaining 8, with dynamic change lower than five-fold, were ranked as Tier 2. Subsequently, two markers in Tier 1 and 2 (CD24 and SUCLG2) failed genomic DNA screening (amplified DNA in addition to mRNA) and were removed. Hence, the process resulted in the final list of 25 Tier 1 and Tier 2 candidate markers (Appendix A). We then combined 23 from the original 29 genes (Appendix A) and 25 new LAMP-compatible markers to form a pool of 48 markers for feature downselection via the selected machine learning approach (Appendix A). See Appendix A for more information.

### 2.3. Selecting Best Alternative Marker Set Using Machine Learning

Constrained by the Myrna cartridge, we needed to downselect from a total of 48 LAMP-compatible targets to 29 genes (plus 3 housekeeping genes). The selection of 29 markers proceeded in two phases:

Phase I: We used a forward selection method [20], a logistic regression (LOGR) model, and a random hyperparameter search to choose the initial set of markers.

Phase II: We used a forward selection method, a multi-layer perceptron model, and Bayesian hyperparameter optimization, with expert judgement to choose the additional markers for a total of 29.

The rationale for this approach and the descriptions of steps are provided in the Appendix A.

### 2.4. Bioinformatics Analysis of the New 29-Marker Set

To understand the biology of the final selected marker set, we used several online tools to investigate the genes both individually and as a group [21]. A complete description of the software tools is in the Appendix A.

For gene set enrichment analyses with Gene Ontology (GO) [22], KEGG [23], and REACTOME [24], we estimated the significance of over-representation of the input genes belonging to a term in the chosen system. In all tests, human transcriptome reference was used but only annotated genes were counted as background in the hyper-geometric test. The *p*-value from the test was then adjusted as false discovery rate (FDR) or q-value using the g:SCS method [25,26] which is a better correction than the Bonferroni adjustment method and the Benjamini–Hochberg correction method. See Appendix A for more information.

For network, upstream regulator, bioprofiler analyses with IPA [27], we included edges represented by various types of relationships including protein-protein interaction, activation, binding, transcription, gene expression correlation, or protein–DNA binding. See Appendix A for more information.

## 3. Results

### 3.1. Selection of LAMP-Optimized the InSep 29-Marker Set

We established training and validation datasets from Inflammatix’s biobank to be roughly balanced across our diagnostic classes of interest (Table 1). In addition to number of samples summarized in Table 1 used for each class in training and validation, patient characteristics and composition were also tabulated for 44 training datasets in Table 2 and Figure 2 validation datasets in Table 3, respectively. Using performance criteria established to select the new LAMP-optimized set of markers (see Methods and Appendix A), we first estimated the ‘baseline’ diagnostic performance metrics for the 29 original markers (columns of A in Table 4).

The number of intended final markers remained 29, given the spaces for 29 markers and 3 housekeeping genes each in duplicates were already allotted in the InSep cartridge design. Having established a basis for performance evaluation, we moved on to the marker swap. Phase I yielded 19 genes listed in Appendix A. Phase II used Bayesian optimization and expert assessment to select additional markers (Figure 1). The process yielded 10 additional markers, for a total of 29 combined with Phase I (Table 5). The diagnostic performance metrics of a neural network classifier developed using the new 29 markers is shown in columns of B in Table 4. Comparison to the results from the original 29 markers shows that the new biomarker set was better or at least noninferior in its overall predictive performance of the InSep bacterial/viral/noninfected classifier, judged by a combination of the clinically relevant metrics. Thus, the goal of the marker swap was achieved, and these 29 markers were “locked” for further development.

### 3.2. Key Biological Functions and Hallmarks of the 29 InSep Genes

The directionality and nominal magnitude of relative changes of the final 29 InSep biomarkers are shown in three comparisons: bacterial infection vs. uninfected control, viral infection vs. uninfected control, and bacterial infection vs. viral infection, together with entrez gene ID, location, and type (Table 5).

Overall, the central themes of the underlying biological functions behind the InSep 29 gene set can be summarized in the following somewhat overlapped hallmarks of human immune response:
Immune cell interaction represented by HLA-DM, CEACAM1, and JUPInflammatory mediators represented by TGFB1, DEFA4, S100A12, and ISG15Cellular metabolism and homeostasis represented by HK3, NMRK1, ZDHHC19, ARG1, CTSB, CTSL, PSMB9, KCNJ2, FURIN, and GADD45AImmune receptors represented by LY86, C3AR1, and CD163Intracellular signaling and antiviral response represented by GNA15, RAPDEG1, PDE4B, BATF, OLFM4, IFI27, and OSAL
and two additional categories: circadian rhythm represented by PER1 [28] and hypoxia stress represented by FAM214A. Importantly, although the above categories are quite broad, nearly every gene has published associations with host response to bacterial or viral infections, infection severity, or inflammation. An interesting side note is that frequently studied targets (e.g., TNF or IL-6) are not represented in the final gene panel because they never ranked high enough in individual sub-analyses for inclusion in our panel [15,16,17]. This is likely because ‘typical’ cytokines are broadly induced by multiple inflammatory pathways, and so may not be informative in a parsimonious signature [15,16,17].

### 3.3. In-Depth Descriptions and Discussion of the Members of the 29 mRNA InSep Set

**ARG1**, arginase 1, catalyzes the hydrolysis of arginine to ornithine and urea. Relevant pathway involvement includes innate immune system, neutrophil degranulation, CDK-mediated phosphorylation and removal of Cdc6, and NgR-p71 (NTR) signaling. ARG1 is strongly upregulated by cytokine signaling and following toll-like receptor stimulation, both of which are important for antimicrobial defense [29].

**BATF**, basic leucine zipper ATF-Like transcription factor, belongs to the AP-1/AFT superfamily of transcription factors that (1) mediate dimerization with members of the Jun family of proteins, (2) control the differentiation of lineage-specific cells in the immune system, and (3) specifically mediate the differentiation of T-helper 17 cells, follicular T-helper cells, CD8(+) dendritic cells, and class-switch recombination (CSR) in B-cells. Other pathways involved in BATF are cytokine signaling and T cell receptor signaling. Functional studies showed that defects in BATF induced multiple defects in T and B cell communication network and significantly impaired antibody responses [30].

**C3AR1**, complement C3a receptor, is a receptor of an anaphylatoxin released during activation of the complement system. Binding of C3a by the encoded receptor activates chemotaxis, granule enzyme release, superoxide production, and bacterial opsonization. Among its related pathways and activities are signaling by G protein-coupled receptor, peptide ligand-binding receptors, innate immune system, complement receptor immune response activity, chemotaxis, and lectin induced complement pathway. Animal models have shown that this gene is important for protection against invasive bacterial infections [31].

**CD163**, also known as macrophage-associated antigen, is a member of the scavenger receptor cysteine-rich (SRCR) superfamily. CD163 is expressed in monocytes and macrophages. It functions as an acute phase-regulated receptor involved in the clearance and endocytosis of hemoglobin/haptoglobin complexes by macrophages, and it thereby protects tissues from free hemoglobin-mediated oxidative damage [32]. It also plays a role in dendritic cells developmental lineage pathway and vesicle-mediated transport pathway. More importantly, it acts as an innate immune sensor for bacteria and inducer of local inflammation [33].

**CEACAM1**, CEA cell adhesion molecular 1, encodes a member of the carcinoembryonic antigen (CEA) gene family, which belongs to the immunoglobulin superfamily. It was found to be a cell-cell adhesion molecule detected in leukocytes, epithelia, and endothelia. It mediates cell adhesion via homophilic as well as heterophilic binding to other proteins of the subgroup. CEACAM1 was shown to be essential for immune synapse formation in CD8+ T cells during infection [34]. It causes negative regulation of T cell mediated cytotoxicity and is upregulated in many bacterial diseases. Receptor upregulation exposes the host to a range of bacterial infections in the respiratory tract [34,35].

**CTSB**, cathepsin B, encodes a member of the C1 family of peptidases. Relevant pathways include: bacterial infections in CF airways, degradation of the extracellular matrix, innate immune system, collagen chain trimerization, neutrophil degranulation, apoptosis and autophagy, and trafficking and processing of endosomal TLRs. CTSB has been shown to interact with bacterial proteins during legionella infection of macrophages, containing infection through programmed cell death [36]. Antigen-presenting cell expression of CTSB has been shown to downregulate Th1 cytokine responses in response to intracellular parasite infection [37].

**CTSL**, Cathepsin L, is a lysosomal enzyme that participates in numerous physiological processes, including apoptosis, antigen processing, and extracellular matrix remodeling, all relevant to host response. Pathological conditions implicated in CTSL are viral or bacterial infection and others such as invasion and metastasis of tumors, atherosclerosis, renal diseases, and diabetes. CTSL expression is important for production of perforin in NK cells and CD8 T cells [38]. The expression of CTSL is upregulated during inflammation and fibrosis [39,40].

**DEFA4**, defensin alpha 4, is one of defensin family members that are antimicrobial and cytotoxic peptides that are involved in host defense and the innate immune system. Abundant in the granules of neutrophils, they are also found in the epithelia. Members of the defensin family are highly similar in protein sequence and distinguished by a conserved cysteine motif. Defensin 4 has important antiviral and antibacterial activity [41,42]. Relative to other defensins, DEFA4 was found to have the highest antibacterial activity against Gram-negative organisms [42].

**FAM214A**, family with sequence similarity 214 member A, has limited annotations. The GWAS catalog includes C-reactive protein measurement and monocyte count. Differential expression of this gene may be related to hypoxia-stress [43].

**FURIN**, also known as proprotein convertase subtilisin/Kevin type 3 (PCSK3), is a member of the subtilisin-like proprotein convertase family, which includes proteases that process protein and peptide precursors trafficking through regulated or constitutive branches of the secretory pathway. FURIN is known to be co-opted by bacteria and viruses to enhance their virulence and spread [44]. It is thought to be one of the proteases responsible for the activation of HIV envelope glycoproteins gp160 and gp140; and may play a role in tumor progression. The spike protein of SARS-CoV-2 can be cleaved by this protease, leading to enhanced viral infectivity [45]. Diseases related to FURIN include avian influenza.

**GADD45A**, growth arrest and DNA damage inducible alpha, is a member of a group of genes whose transcript level are increased following stressful growth arrest conditions and treatment with DNA-damaging agents, as well as other cellular stress. The encoded protein responds to environmental stress by mediating activation of the p38/JNK pathway via MTK1/MEKK4 kinase. The DNA damage-induced transcription of this gene is mediated by both p53-dependent and -independent mechanisms. GADD45A is important for neutrophil and macrophage function in response to lipopolysaccharide (LPS), a bacterial toxin, specifically modulates innate immune functions of granulocytes and macrophages by differential regulation of p38 and JNK signaling [46].

**GNA15**, G-protein subunit alpha 15, is involved in GPCR super pathway. Noticeably, the GWAS catalog for GNA15 gene are all relevant: counts of myeloid white cells, neutrophils, leukocytes, eosinophils, and monocytes. Macrophage GNA15 is involved with immune signaling related to hematopoiesis and inflammatory response [47].

**HK3**, hexokinase 3, is an enzyme that catalyzes the phosphorylation of hexose, such as D-glucose and D-fructose, to hexose 6-phosphate, the first step in most glucose metabolism pathways. It plays a role in innate immune system and neutrophil degranulation in addition to galactose metabolism and glucose metabolism. Hexokinases have been shown to serve as innate immune receptors for bacterial cell wall peptidoglycan [48] and are upregulated during mycobacterial infections [49].

**HLA-DMB**, major histocompatibility complex, class II, DM beta, belongs to the HLA class II beta chain paralogues. This class II molecule is a heterodimer consisting of an alpha (DMA) and a beta (DMB) chain, both anchored in the membrane. It is located in intracellular vesicles. DM plays a central role in the peptide loading of MHC class II molecules by helping to release the class II-associated invariant chain peptide (CLIP) molecule from the peptide binding site. Class II molecules are expressed in antigen presenting cells (APC): B lymphocytes, dendritic cells, macrophages. In addition to its important role in antigen presentation, HLA-DMB has been shown to decrease viral protein expression through modulation of the autophagosome [50].

**IFI27**, interferon alpha inducible protein 27, plays a critical role in induction of cell apoptosis and known for an antiviral activity. Upregulated by type I interferons, it is involved in innate immune response and interferon signaling in immune system. Interferon alpha is a principal component of antiviral host defense, inducing cell signaling and subsequent production of several antiviral proteins [51]. IFI27 may also play a role in the vascular response to injury. In the innate immune response, it is known to have antiviral activity against a number of viruses [52,53,54,55]. Upregulation of IFI27 has been observed after Toll-like receptor (TLR) 7 stimulation and occurred primarily in plasmacytoid dendritic cells and NK cells. TLR7 is the innate immune receptor for single stranded RNA, a common feature of many viral genomes. In respiratory infections, IFI27 expression has been shown to discriminate bacterial from viral illness [55].

**ISG15**, ISG15 ubiquitin like modifier, is also known as interferon-induced 17 KDa protein (IP17) or ubiquitin cross-reactive protein (UCRP). The protein encoded by this gene is conjugated to intracellular target proteins upon activation by interferon-alpha and interferon-beta. Its functions are relevant–chemotactic activity towards neutrophils, direction of ligated target proteins to intermediate filaments, cell-to-cell signaling, and antiviral activity during viral infections. It plays a key role in the innate immune response to viral infection either via its conjugation to a target protein (ISGylation) or via its action as a free or unconjugated protein. ISGylation involves a cascade of enzymatic reactions involving E1, E2, and E3 enzymes which catalyze the conjugation of ISG15 to a lysine residue in the target protein. It exhibits antiviral activity towards both DNA and RNA viruses. The secreted form of ISG15 can induce natural killer cell proliferation, act as a chemotactic factor for neutrophils, and act as a IFN-gamma-inducing cytokine playing an essential role in antimycobacterial immunity. The secreted form acts through the integrin ITGAL/ITGB2 receptor to initiate activation of SRC family tyrosine kinases including LYN, HCK, and FGR which leads to secretion of IFNG and IL10; the interaction is mediated by ITGAL [56].

**JUP**, junction plakoglobin, encodes a major cytoplasmic protein which is the only known constituent common to sub-membranous plaques of both desmosomes and intermediate junctions. It plays a critical role in the arrangement and function of the cytoskeleton and the cells within the tissue. Its functions include innate immune response, cell junction organization, and cell adhesion. Plakoglobin expression has been shown to be affected by exposure to inflammatory cytokines and viral proteins [57,58].

**KCNJ2**, potassium inwardly rectifying channel subfamily J member 2, encodes an integral membrane protein, inward-rectifier type potassium channel. This gene is upregulated during tissue repair, including in fibrotic lung and heart disease [59]. KCNJ2 modulates cell growth and apoptosis, and overexpression has been associated with inflammatory cytokine expression [60]. Furthermore, KCNJ2 upregulation has been shown to be a biomarker of cardiomyocyte death and hypoxia [61,62].

**LY86**, lymphocyte antigen 86, also known as myeloid differentiation-1 (MD-1), is involved in innate immune system, and activated TLR4 signaling. LY86 has been shown to cooperate with CD180 and TLR4 to mediate the innate immune response to bacterial LPS [63]. Expression of this gene has also been shown to be important for preventing pathological cardiac remodeling after injury [64].

**NMRK1**, previously known as C9orf95, is nicotinamide riboside kinase 1. NMRK1 is important for nicotinamide adenine dinucleotide (NAD) metabolism and metabolism of water-soluble vitamins and cofactors. Expression of this gene has been shown to be upregulated in virally infected cells, including those infected with SARS-CoV-2 [65].

**OASL**, 2′-5′-oligoadebylate synthetase like, is also known as TR-interacting protein 15 (TRIP15). OASL is an interferon stimulated gene (ISG) that is directly and rapidly induced upon viral infection [66]. It complexes with cellular RNA and DNA sensors and acts as a molecular rheostat for modulating interferon response to RNA and DNA virus infection [67]. Among its related pathways are innate immune system, interferon alpha/beta signaling, interferon gamma signaling, cytokine signaling in immune system, immune response, and PI3K-Akt signaling pathway. OASL is important for antiviral innate immunity, helping overcome viral evasion by RNA viruses [68].

**OLFM4**, olfactomedin 4, is also known as antiapoptotic protein GW112. Initially cloned from human myeloblasts, this gene is found to be selectively expressed in inflamed colonic epithelium. The encoded protein is an antiapoptotic factor that promotes tumor growth and is an extracellular matrix glycoprotein that facilitates cell adhesion [69]. Among its related pathways are innate immunity, cell adhesion, and neutrophil degranulation. It may also promote proliferation of pancreatic cancel cells by favoring the transition from the S to G2/M phase. Expression of OLFM4 has been associated with respiratory viral infection severity in children [70].

**PDE4B**, phosphodiesterase 4B, is a member of the type IV, cyclic AMP-specific, cyclic nucleotide phosphodiesterase (PDE) family. The encoded protein regulates the cellular concentrations of cyclic nucleotides and thereby play a role in signal transduction. Among its related pathways are signaling by GPCR and myometrial relaxation and contraction pathways, as well as T cell activation [71].

**PER1**, period circadian regulator 1, is one of a few well-studied circadian regulators. It is expressed in a circadian pattern in the suprachiasmatic nucleus as the primary circadian pacemaker in the mammalian brain. PER1 has also been shown to have an important role in limiting the excessive innate immune response to bacterial LPS [28].

**PSMB8**, proteasome 20S subunit beta 8, is an essential component of the immunoproteasome, needed for the processing of class I major histocompatibility complex (MHC) peptides. Located in the class II region of the MHC, the expression of PSMB8 is induced by gamma interferon [72]. It is important for T cell antigen presentation, especially for viral antigens [73]. PSMB8 also plays a role in apoptosis via the degradation of the apoptotic inhibitor MCL1.

**RAPGEF1**, rap guanine nucleotide exchange factor 1, encodes a human guanine nucleotide exchange factor. Among related pathways include common cytokine receptor gamma-chain family signaling pathways and integrin-mediated cell adhesion HGF pathway. It transduces signals from CRK by binding the SH3 domain of CRK and activating several members of the Ras family of GTPases. This gene is involved in apoptosis, integrin-mediated signal transduction, and cell differentiation [74].

**S100A12**, S100 calcium binding protein A12, also known as neutrophil S100 protein, is a calcium-, zinc-, and copper-binding protein which plays a prominent role in the regulation of inflammatory processes and immune response. Its proinflammatory activity involves recruitment of leukocytes, promotion of cytokine and chemokine production, and regulation of leukocyte adhesion and migration. It acts as a monocyte and mast cell chemoattractant. It can stimulate mast cell degranulation and activation which generates chemokines, histamine, and cytokines inducing further leukocyte recruitment to the sites of inflammation. Furthermore, it can inhibit the activity of matrix metalloproteinases: MMP2, MMP4, and MMP9 by chelating Zn(2+) from their active sites. S100 proteins are localized in the cytoplasm and/or nucleus of a wide range of cells and is also involved in the regulation of a number of cellular processed such as cell cycle progression and differentiation. Relevantly, the protein is involved in specific calcium-dependent signal transduction pathways and its regulatory effect on cytoskeletal components may modulate various neutrophil activities. The protein includes an antimicrobial peptide which has antibacterial activity [75].

**TGFBI**, transforming growth factor beta induced, encodes an RGD-containing protein that binds to type I, II, and IV collagens. The RGD motif is found in many extracellular matrix proteins modulating cell adhesion and serves as a ligand recognition sequence for several integrins. This protein plays a role in cell-collagen interactions and may be involved in endochondral bone formation in cartilage. The protein is induced by transforming growth factor-beta and other cytokines, and acts to inhibit white blood cell adhesion [76], including after macrophage ingestion of apoptotic cells [77].

**ZDHHC19**, zinc finger DHHC-type palmitoyltrasferase 19, is known to be associated with some viral infections. Its annotations include protein-cysteine S-palmitoyltrasferase activity and mediation of palmitoylation of RRAS, leading to increased cell viability. A major immune function of ZDHHC19 is to facilitate activation of STAT3 upon cytokine stimulation [78], which is important for immune signaling in infection and inflammation [79]. Interestingly, GWAS studies identified several phenotypes associated with ZDHHC19 variants include erythrocyte count, red blood cell distribution width and mean corpuscular hemoglobin concentration. Expression of ZDHHC19 has been associated with sepsis and septic shock in previous human studies [80].

### 3.4. Results from Pathway Enrichment Analyses of The InSep 29-mRNA Set

The final 29 biomarkers in Table 5 were subjected to multiple gene set enrichment analyses using various methods: GO terms of biological process (BP), cellular compartment (CC), and molecular function (MF); pathways in KEGG; and reactions in REACTOME (summaries of these techniques and databases are in the Appendix A). Figure 2 summarizes results from this analysis with FDR < 0.01 in the first five columns. It also illustrates the membership of these significant terms in our 29 genes (the last 29 columns in Figure 2) as heatmap.

#### 3.4.1. InSep 29 mRNAs in Gene Ontology

For GO BP, we found that 28 terms enriched significantly, and the majority were associated with immune and inflammatory responses such as: (1) immune effector process; (2) leukocyte, neutrophil, myeloid cell, granulocyte, and cell activation in involved in immune response; (3) neutrophil and leukocyte degranulation; and (4) cellular response to organic substance and chemical stimulus. Other processes such as exocytosis play a role in transporting vesicles. First, 11 of the 29 genes (CEACAM1, ARG1, DEFA4, S100A12, C3AR1, CTSB, JUP, OLFM4, HK3, HLA-DMB, and BATF) are commonly involved in four GO BPs: (1) immune response, (2) immune effector process, (3) leukocyte activation involved in immune response, and (4) cell activation involved in immune response. These pathway terms have members largely overlapped (see heatmap in Figure 2). Unlike CEACAM1, C3AR1, OLDM4, and HK4 that are mainly involved in neutrophil activities; ARG1, DEFA4, JUP, S100A1, and CTSB are also known to play roles in other biological processes relevant to host response such as pathogen defense (ARG1, DEFA4, S100A12) and response to type I interferon (JUP). Second, IFI27, OASL, and ISG15 are all involved in viral life cycle, regulation of symbiosis, encompassing mutualism through parasitism, regulation of viral genome replication, regulation of striated muscle contraction, type I interferon signaling pathway, and response to type I interferon. Third, the contributing genes to regulation of immune effector process include ARG1, CEACAM1, C3AR1, HLA-DMB, and RAPGEF1.

Overall, the GO BP term analysis of the 29 InSep genes revealed major biological processes relevant to its diagnostic purpose–via immune response to bacterial or viral invasion by host cells. The number of GO BP terms to which a gene belongs can be very high for some well-known and extensively studied genes (e.g., 246 for CTSB and 224 for ARG1) or very low for others, depending on the omnibus or unique role of these genes can play or simply the biases of our genome research [82], an interesting topic beyond the scope of this work.

Significant terms in GO CC include granule and vesicle (specific and secretory) and their lumen for their obvious roles in host response. Extracellular vesicles (EV), the most common exosomes, consistently produced by viral-infected cells, play crucial roles in mediating communication between infected and uninfected cells. Recent studies [83,84] have revealed pathophysiological roles for EV in various viral infections, including human immunodeficiency virus, coronavirus, and human adenoviruses. Critically, viruses can exploit EV formation, secretion, and release pathways to promote infection, transmission, and intercellular spread [24]. Consequently, EV production has been investigated as a potential tool for the development of improved viral infection diagnostics and therapeutics. In a recent review on EV–virus relationships, Ipinmoroti and Matthews [85] summarized the roles of EVs in pathophysiological pathways, immunomodulatory mechanisms, and utility for biomarker discovery. They also discussed the potential for EVs to be exploited as diagnostic and treatment tools for viral infection.

For GO MF terms, no significant terms were found in the enrichment analysis; and the top one with FDR of 0.06 is collagen binding due to their three member CTSB, CTSL, and TGFBI.

#### 3.4.2. InSep 29 mRNAs in KEGG Pathways

In the KEGG analysis, we found one significant pathway for our 29 gene set, *antigen processing and presentation,* due to three member genes: CTSB, CTSL, and HLA-DMB via their roles in MHC II pathway that leads to CD4 T cell receptor signaling pathway for cytokine production and activation of other immune cells. It is known that the members of antigen processing and presentation overlap with those of the T cell exhaustion signaling pathway, MSP-RON signaling in macrophages pathway, and the role of NFAT in regulation of the immune response.

The second-ranked pathway (though non-significant) was *Staphylococcus aureus* infection, represented by DEFA4, C3AR1, and HLA-DMB. Next is apoptosis–Cathepsin B and L, together with GADD45A are involved in apoptosis among other processes. Moreover, HLA-DMB, GADD45A, and ISG15 are reported to be involved in Epstein–Barr virus infection, one of the most common human viruses in the world, causing infectious mononucleosis and other illnesses. Additionally, autophagy is found more in viral than in bacterial infection, likely because a subset of viruses and bacteria subvert the autophagic pathway to promote their own replication [86].

#### 3.4.3. The InSep 29 mRNAs in Reactome

Three terms in Reactome analysis were found significant (all with FDR < 0.0005): immune system, innate immune system, and neutrophil degranulation due to contributions of 18, 13, and 9 members respectively from the 29 input genes as shown in Figure 2. Marginally significant ones (not shown) with contributing members ranging from 2 to 4 include trafficking and processing of endosomal toll-like receptor (TLR), collagen degradation, TLR cascades, interferon alpha/beta signaling, CD163 mediating anti-inflammatory response, cytokine signaling in immune system, MHC class II antigen presentation, degradation of the extracellular matrix, extracellular matrix organization, interferon signaling, all relevant to immune response.

### 3.5. Networks and Subnetworks Involving the InSep 29 mRNAs

With the 29 InSep genes as input seed nodes using knowledge base captured in the IPA system for humans, we recovered three networks labelled A–C as shown in Figure 3.

Network A has a total of 35 nodes; of the 29 InSep genes, 20 are found in this network, either over-expressed (red) or under-expressed (green) in viral infection in comparison with uninfected controls. Top diseases and functions involved in this network include immunological disease, inflammatory response, and organism injury and abnormalities. The first one is the MHC class II molecules, found on antigen-presenting cells. As MHC class II protein complex is encoded by the human leukocyte antigen complex (HLA), HLA-DMB is under-expressed in both viral and bacterial infection, but by a greater magnitude in bacterial than viral infection. CSTL(Cathepsin L) is involved in MHC-mediated antigen processing and presentation, also shown in the network. In addition, CTSL plays a major role in intracellular protein catabolism. Its substrates include collagen and elastin, as well as alpha-1 protease inhibitor, a major controlling element of neutrophil elastase activity. CTSL is also known to be associated with Middle East Respiratory Syndrome and bacterial infections in CF airways. The second key driver is IFN beta as well as IFN alpha (interferon type I) for upregulating the expression of IFI27, with greater magnitude in viral than in bacterial infection; type I interferons are produced by fibroblasts and monocytes when a viral infection is recognized [87]. C3AR1, Complement C3a Receptor 1, is impacted by several pathways and is upregulated by MHC class II, type I interferons, interleukin 12 complex, immunoglobulin G, and TNF via various mechanisms.

Network B involves 6 of the 29 InSep markers with 3 upregulated (red) and 3 downregulated (green) among its 35 nodes. Top diseases and functions of them are involved in cardiac arrythmia, cardiovascular system development and function in general, and organism development via cellular matrix. TGFBI, transforming growth factor beta induced, is associated with many diseases via its pathway involvement in metabolism of proteins, Wnt, Hedgehog, and Notch signaling. It plays an important role in cell adhesion, cell-collagen interaction, and extracellular matrix binding. BATF, basic leucine zipper ATF-like transcription factor, regulates expression of multiple interleukins, IL2, IL4, IL5, IL10, and IL13, proliferates CD8+ T lymphocytes, and plays a role in cytokine signaling in immune system.

Network C has only six nodes but includes a key gene from the 29 markers, DEFA4, and a close family member, DEFA1 (also known as HNP-1). These defensins, a family of antimicrobial and cytotoxic peptides, are known to be involved in host defense and are abundant in the granules of neutrophils, localizing with pro-inflammatory cytokines. Top diseases and functions associated with this network include drug metabolism, molecular transport, and tissue morphology. Together, this network reveals the relevant biology of DEFA4 as a biomarker for bacterial vs. viral infection as well as for severity.

One can grow a network by connecting additional nodes to the existing nodes already in the networks according to their relationships such as protein–protein interaction, activation, binding, transcription, gene expression correlation, or protein–DNA binding. As an example, in Figure 3 we expanded the network in (A) by connecting more genes to the genes of interest, resulting in the final network in (D). Interestingly, some genes (ISG15, PSMB9, CEACAM1, CADD45A) are highly interconnected based on current knowledge, and others (OASL, PDE4B, BATF, GNA15, HK3) have fewer connections. For example, ISG15 is regulated by many including STAT1, STAT2, TP53, HDAC4, HDAC11, IRF3, and IRF8. No extension was found for three genes: LY86, OLFM4, and FAM214A. However, OLFM4 has recently been tied to sepsis severity [88], meaning the lack of nodes may simply reflect poor knowledge in some networks.

### 3.6. Upstream Regulators Impacting the InSep 29 mRNAs

The upstream regulator analysis using the IPA tool takes into account the directionality and magnitude changes of 29 input genes. We used effect size from bacterial infection vs. uninfected control (Table 5) to show a regulator network derived for bacterial infection in a cellular view in Figure 4. Evidently, the effect on impacting expression of many biomarkers were found to be mediated through a few key drivers such as TNF (tumor necrosis factor), a cytokine that contributes to the acute phase reaction, and modulates type 1 immune responses to infections [89]. Interleukin 6 (IL-6) is not in our gene set but connects to several of genes either upstream or downstream (Figure 4), and also acts as both a pro-inflammatory cytokine and an anti-inflammatory myokine [90]. Smooth muscle cells in the tunica media of many blood vessels also produce IL-6 as a pro-inflammatory cytokine. Conversely, TGM2 mediates effect directly on targets DEFA4, OLFM4, HK4, and indirectly on other biomarkers via other targets.

### 3.7. Connection of 29 mRNAs to Relevant Phenotypes

The knowledge base in IPA also allowed us to link the 29 markers to phenotypes. Here we highlight such connections in four subnetworks in Figure 5, showing their connections to (A) cell proliferation of T lymphocytes, (B) organismal death, (C) infection by RNA virus, and (D) viral infection. Figure 5A shows 5 genes (ARG1, BATF, CEACAN1, GADD45A, and HLA-DMB) linking to cell proliferation of T lymphocytes, either via activation or inhibition. Presumably, these five mRNAs provide the InSep algorithms with a partial measure of T lymphocyte proliferation as part of host response to infection. We found 11 genes to be connected to organismal death (Figure 5B), an important phenotype but tested to be insignificant in enrichment analyses due to relatively large number of genes whose annotations share the term ‘organismal death’. Respectively 13 and 16 genes were found to be linked to infection by RNA virus and viral infection through various mechanisms (Figure 5C,D).

## 4. Discussion

We here show an effective optimization of our 29-mRNA panel to allow for transition to a rapid qRT-LAMP-based test, and provide a biological understanding of the final panel. During the assay development process and a new generation classifier training, 13 of the original 29 genes were replaced in the final locked-in model for InSep due to the model optimization primarily for isothermal amplification process used on our chosen platform of loop-mediated isothermal amplification (LAMP). This list of 13 genes: C11orf74, CIT, GPAA1, HIF1A, HLA-DPB1, LAX1, MTCH1, OR52R1, RGS1, RPDRIP1, SEPP1, TNIP1, and TST were replaced collectively by a new list of 13 other genes: ARG1, C9orf95 (NMRK1), CTSL1, FURIN, GADD45A, HLA-DMB, ISG15, OASL, OLFM4, PDE4B, PSMB9, RAPDEF1, and S100A12. We show that the classifiers based on the final 29 markers produce an overall equal or better performance than the classifiers based on the previous 29 markers, but now can be measured using a rapid assay (qRT-LAMP) to fit into workflow [14].

The 29 biomarkers were selected based on their analytical performance with qRT-LAMP and their significance in predicting the classes to be tested: viral infection, bacterial infection, or 30-day mortality. In building for clinical effectiveness, our data-driven approach chose genes which we have here shown to be biologically plausible as highly linked to relevant immune functions. While some genes are not well-studied in infections yet, this is likely due to the ‘streetlamp effect’ of biological research–well-studied genes become more and more well-studied, and pathways/knowledge databases only reflect known associations [82]. Thus, for instance, OLFM4 was included strictly because of its apparent ability to estimate sepsis severity, but knowledge bases may not yet have incorporated its early linkage with sepsis [88].

There are limitations in this type of biological interpretation research in general. First, the analysis depends on annotations found in the databases that are available, such as gene set membership in pathways, pathway topology, presence of genes in certain biological processes, and the backbone network constructed based on available literature. These annotations and knowledge base in general, however, are far from being complete and have highly variable degrees of reliability. In addition, many terms for molecular functions or biological processes are general, deprived of cell type, compartment, or developmental context. Therefore, interpretation of pathway analysis results for biomarkers as those reported in this work should be used with caution. Second, the knowledge base upon which we rely for the analysis is highly biased–we know what we know, but are not able to make connections based on yet-undiscovered biology [82]. That said, this work not only helped us to gain an initial understanding of biological processes, pathway participation, and cellularity switches reflected by the transcriptional changes we measure in the 29 InSep biomarkers, but also will help us to design and conduct targeted studies to further elucidate their individual and coordinated roles in reporting bacterial and viral infection together with its severity in the future.

It should be emphasized that here we are not pursuing biomarker discovery for biological mechanisms underlying a biological condition or disease state, as is the case in mechanistic studies. Rather, we are developing a diagnostic and prognostic test for acute infections and sepsis with the 29 biomarkers using a data-driven approach [18,91]. Namely, we rely on the readout of the 29 mRNAs to collectively provide a molecular portrait of the physiological state of the immune response for purposes of clinical diagnostics and prognostics. The classifier, trained by machine learning algorithms, integrates the measurements of all 29 mRNAs in its totality and outputs well-calibrated scores to help clinicians to take the correct clinical actions for patients in need, as discussed by Ducharme et al. [13]. In the long run, rapid host-response diagnostics may also prove valuable for identifying other infection types, such as fungal or malarial infections.

## 5. Conclusions

Diagnostic or prognostic tests based on multi-gene signatures must be both clinically effective and biologically significant in order to find broad clinical use. We here optimized a 29-mRNA set for rapid measurement using qRT-LAMP on the disposable, cartridge-based Myrna platform. We further demonstrated the biological plausibility of the final chosen 29-mRNA set, with multiple linkages at the gene, pathway, and network level to leukocyte biology and infection response. A prospective study of the rapid LAMP-based 29-mRNA panel is underway as part of a registrational trial.

## Figures and Tables

**Figure 1 jpm-11-00735-f001:**
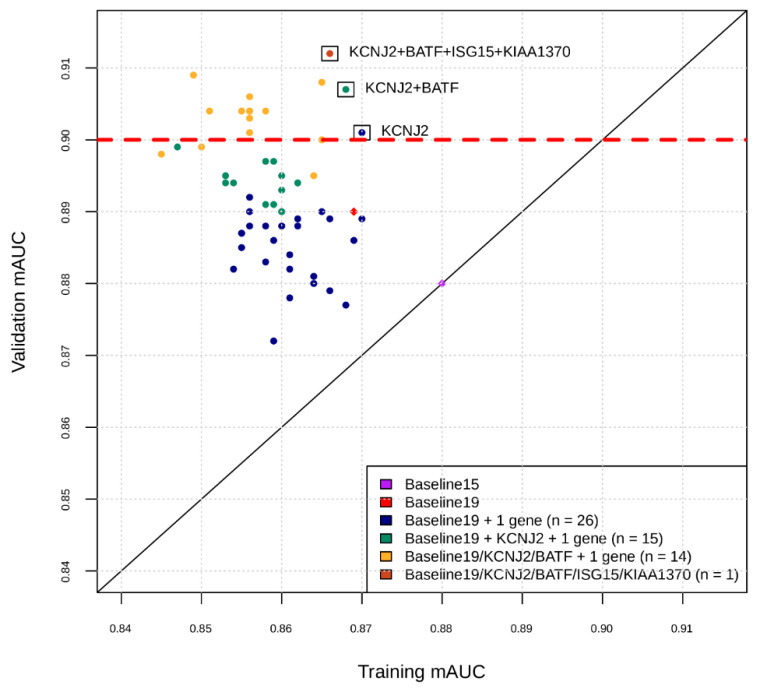
Intermediate snapshot of the Phase II of the marker selection by machine learning. *X*-axis is the cross-validation AUC in training for best model found by Bayesian Hyperparameter Optimization using features comprising current marker set plus one marker at a time. *Y*-axis is the AUC of that model applied to validation set. For example, the blue dots represent training and validation AUCs for feature sets consisting of the 19 markers found in Phase I, plus one of the markers in the remaining set of markers. KCNJ2 was added to current marker set and the process repeated for the remaining set of markers (green dots), etc.

**Figure 2 jpm-11-00735-f002:**
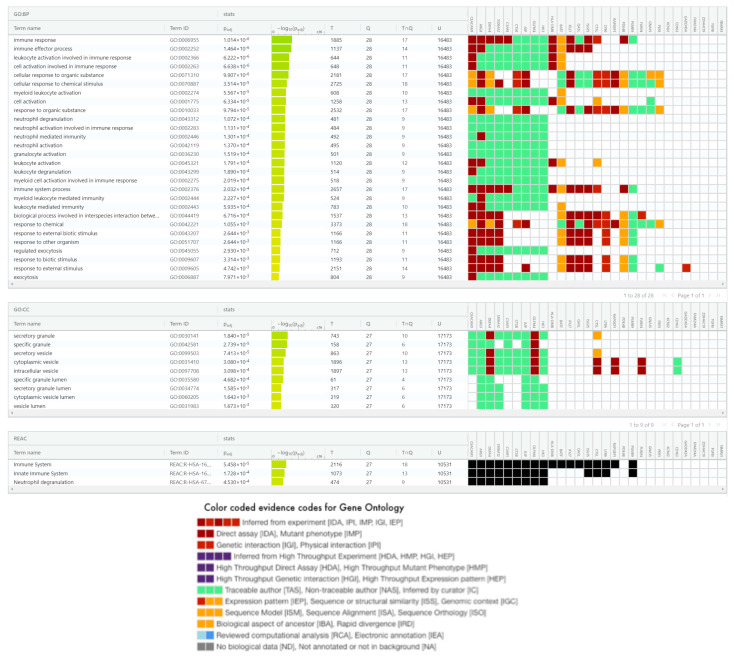
Significant biological process (28) and cellular compartment (9) terms in gene ontology, and reactome (3) terms found for 29 InSep genes. Terms in GO BP, GO CC, and Reactome are ordered respectively by their adjusted *p*-values in the enrichment analysis using g:Profler [25,26,81]. Also shown are members of each term found in the 29 genes with color coded evidence code legend below the table.

**Figure 3 jpm-11-00735-f003:**
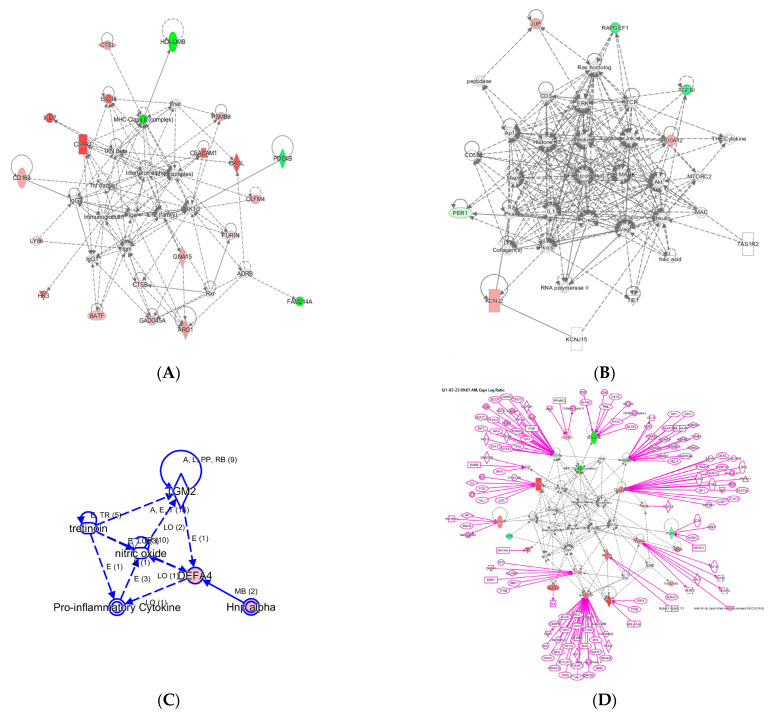
Three networks built with the 29 InSep genes as input seed nodes (**A**–**C**). The directionality of relative changes for each gene (red for over-expressed and green for under-expressed) are referred to the comparison of viral infection versus uninfected control. (**D**) is an extended network grown on network (**A**). The added part was shown as purple color.

**Figure 4 jpm-11-00735-f004:**
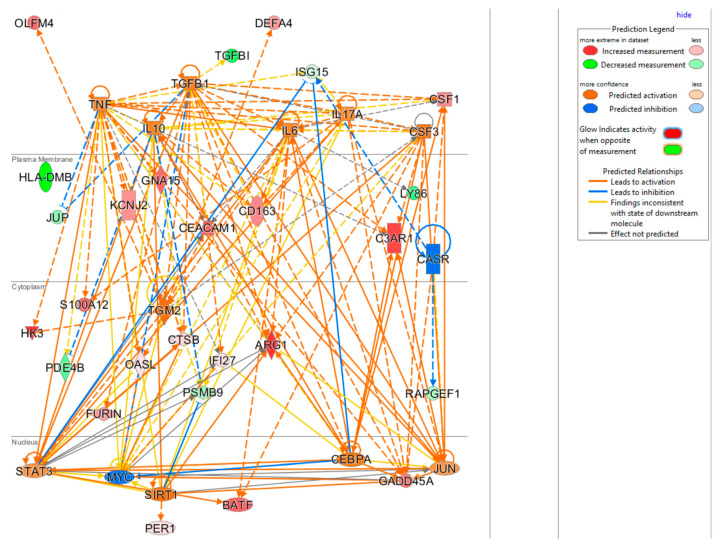
Cellular view of a regulator network for 29 input genes. This network was derived based on the knowledge base in IPA with InSep 29 markers as input for bacterial infection.

**Figure 5 jpm-11-00735-f005:**
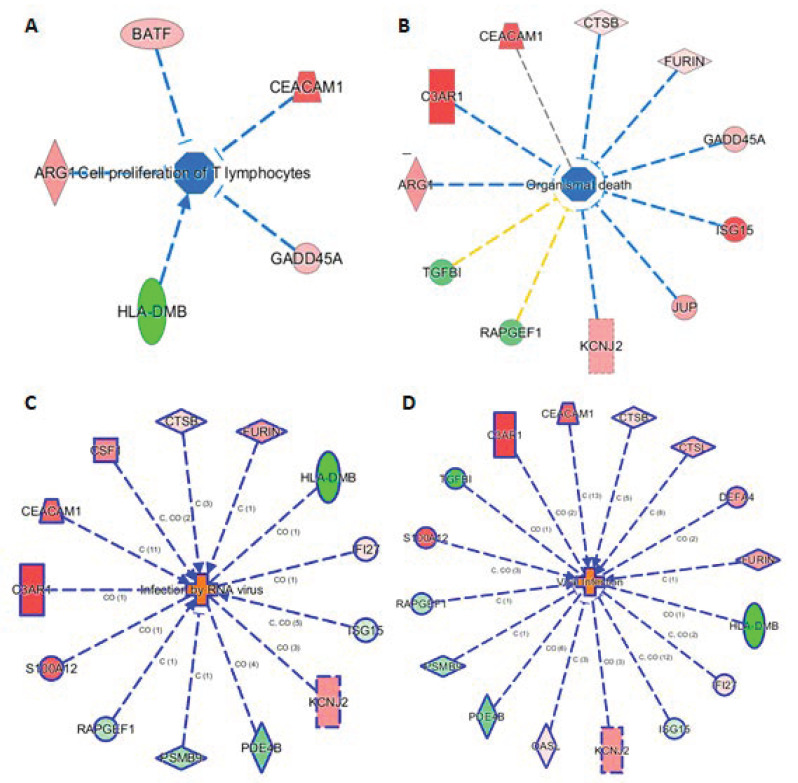
Subnetworks for selected InSep biomarkers connecting to phenotypes of interest. Disease-related phenotypes are (**A**) cell proliferation of T lymphocytes, (**B**) organismal death, (**C**) infection by RNA virus, and (**D**) viral infection.

**Table 1 jpm-11-00735-t001:** Number of samples used in datasets for training and validation to assess the performance of the original and swapped marker sets.

Dataset	Class		
	Bacterial	Viral	Noninfected
**Training**	1028	1049	1082
**Validation**	240	119	18

**Table 2 jpm-11-00735-t002:** Patient characteristics and composition of datasets used in training for the marker swap optimization. ID = Infectious Disease; COPD = Chronic Obstructive Pulmonary Disease; ICU = Intensive Care Unit; CAP = Community-Acquired Pneumonia; SIRS = Systemic Inflammatory Response Syndrome; TB = Tuberculosis; SJIA = Systemic Juvenile Idiopathic Arthritis; HRV = Human Rhinovirus; RSV = Respiratory Syncytial Virus.

Study	First Author	Description ^a^	N	Median Age (IQR)	Female (%) ^b^	Platform	Country	Bacteria (%)	Virus (%)	Noninfected (%)
E-MEXP-3589	Almansa	Patients hospitalized with COPD exacerbation	23	70.1	5 (22)	Agilent	Spain	4 (17)	5 (22)	14 (61)
E-MTAB-1548	Almansa	Surgical patients with sepsis (EXPRESS)	140	72 (61–78)	44 (31)	Agilent	Spain	82 (59)	0	58 (41)
E-MTAB-3162	van de Weg	Patients with dengue	21	20 (17–28)	10 (48)	Affymetrix	Indonesia	0	21 (100)	0
E-MTAB-5273/5274	Burnham	Sepsis due to faecal peritonitis or pneumonia	227	69 (54–77)	99 (44)	Illumina	UK	227 (100)	0	0
E-MTAB-5638	Almansa	ICU patients w/ventilator-associated pneumonia	17	68 (± 26)	7 (41)	Agilent	Spain	0	0	17 (100)
GlueBuffyHCSS ^c^	Multiple	Trauma patients	320	33 (25–43)	43 (36)	Affymetrix	USA	46 (14)	0	274 (86)
GSE13015 (GPL6102)	Pankla	Sepsis, many cases from burkholderia	45	54 (48–61)	19 (42)	Illumina	Thailand	45 (100)	0	0
GSE13015 (GPL6947)	Pankla	Sepsis, many cases from burkholderia	15	49 (44–60)	9 (60)	Illumina	Thailand	15 (100)	0	0
GSE21802	Bermejo-Martin	Pandemic H1N1 in ICU	10	unknown	unknown	Illumina	Canada	0	10 (100)	0
GSE29385	Naim	Patients with influenza and other respiratory infections	80	unknown	unknown	Illumina	Vietnam	0	80 (100)	0
GSE61821	Hoang	Febrile patients positive for H1N1, H3N2	48	40 (20–51)	unknown	Illumina	Vietnam	0	48 (100)	0
GSE42026	Herberg	Children with H1N1/09, RSV or bacterial infection	59	unknown	26 (44)	Illumina	UK	18 (31)	41 (69)	0
GSE72810	Herberg	Febrile children with bacterial or viral infection	15	unknown	7 (47)	Illumina	UK	5 (33)	10 (67)	0
GSE103842	Rodriguez-Fernandez	Children with RSV infection	62	3 (2–5.3)	23 (37)	Illumina	USA	0	62 (100)	0
GSE77087	de Steenhuijsen Piters	Children with RSV infection	41	5.4 (1.7–8.3)	16 (39)	Illumina	USA	0	41 (100)	0
GSE22098	Berry	Patients with active TB and other IDs	193	16 (11–26)	134 (69)	Illumina	UK	52 (27)	0	141 (73)
GSE30119	Banchereau	Patients w/*Staphylococcus aureus* infection	59	6.5 (2–11)	25 (42)	Illumina	USA	59 (100)	0	0
GSE25504 (GPL13667)	Smith	Neonatal sepsis	12	0	4 (33)	Affymetrix	UK	9 (75)	3 (25)	0
GSE25504 (GPL6947)	Smith	Neonatal sepsis	21	0	10 (48)	Illumina	UK	20 (95)	1 (5)	0
GSE27131	Berdal	Severe H1N1	7	38 (33–50)	1 (14)	Affymetrix	Norway	0	7 (100)	0
GSE28750	Sutherland	Sepsis, post-surgical SIRS	21	unknown	10 (48)	Affymetrix	Australia	10 (48)	0	11 (52)
GSE28991	Naim	Acute dengue	11	unknown	unknown	Illumina	unknown	0	11 (100)	0
GSE32707	Dolinay	Critically ill patients in Brigham\& Women’s ICU	44	56 (45–59)	8 (18) ^b^	Illumina	USA	0	0	44 (100)
GSE40012	Parnell	Bacterial or influenza A pneumonia or SIRS	36	59 (46.5–67)	16 (44)	Illumina	Australia	16 (45)	8 (22)	12 (33)
GSE40165	Nguyen	Children and adolescents with dengue	123	12 (10–14)	38 (31)	Illumina	Vietnam	0	123 (100)	0
GSE40396	Hu	Febrile young children	30	1 (0.3–1.6)	13 (43)	Illumina	USA	8 (27)	22 (73)	0
GSE40586	Lill	Community-acquired bacterial meningitis	15	57 (53–71)	unknown	Affymetrix	Estonia	15 (100)	0	0
GSE42834	Bloom	Bacterial pneumonia or sarcoidosis	82	unknown	40 (49)	Illumina	UK, France	14 (17)	0	68 (83)
GSE47655	Stone	Acute anaphylaxis	6	unknown	unknown	Affymetrix	Australia	0	0	6 (100)
GSE51808	Kwissa	Dengue patients	28	unknown	unknown	Affymetrix	Thailand	0	28 (100)	0
GSE57065	Cazalis	Septic shock	28	62 (54–76)	9 (32)	Affymetrix	France	28 (100)	0	0
GSE57183	Senoi	SJIA patients	11	4 (3–7)	6 (55)	Illumina	USA	0	0	11 (100)
GSE60244	Suarez	Lower respiratory tract infections	93	63 (50–77)	56 (60)	Illumina	USA	22 (24)	71 (76)	0
GSE63881	Hoang	Kawasaki disease	171	3 (1.4–4.3)	69 (40)	Illumina	USA	0	0	171 (100)
GSE64456	Mahajan	Febrile infants (60 days of age and younger)	200	0.1 (0.06–0.13)	94 (47)	Illumina	USA	89 (44)	111 (56)	0
GSE65682	Scicluna	Suspected but negative for CAP	33	59 (48–67)	11 (33)	Affymetrix	Netherlands	0	0	33 (100)
GSE66099	Sweeney	Pediatric ICU (sepsis, septic shock, or SIRS)	150	2.5 (1–3)	56 (37)	Affymetrix	USA	109 (73)	11 (7)	30 (20)
GSE67059	Heinonen	Children with HRV infection	80	0.8 (0.3–1.3)	27 (34)	Illumina	USA	0	80 (100)	0
GSE68310	Zhai	Outpatients with acute respiratory viral infections	104	21 (20–23)	54 (52)	Illumina	USA	0	104 (100)	0
GSE69528	Khaenam	Sepsis, many cases from burkholderia	83	unknown	44 (53)	Illumina	Thailand	83 (100)	0	0
GSE73461	Wright	Children with various IDs	308	3 (1–9)	143 (46)	Illumina	UK	52 (17)	94 (31)	162 (52)
GSE77791	Plassais	Severe burn shock	30	48 (40–55)	9 (30)	Affymetrix	France	0	0	30 (100)
GSE82050	Tang	Moderate and severe influenza infection	24	65 (49–74)	10 (42)	Agilent	Germany	0	24 (100)	0
GSE111368	Dunning	Adults hospitalized with influenza	33	38 (29–49)	18 (55)	Illumina	UK	0	33 (100)	0

Notes for Table 2: ^a^ Study description is taken from the study’s corresponding publication and includes some patients that were excluded from the training set. ^b^ Numbers and percentages shown reflect the fact that some patients in the study had unknown/unreported sex. ^c^ Total study sample size (N) and statistics of bacterial infection and non-infected composition are based on 320 patient samples used for marker swap (including temporal replicates from non-infected patients) while age and sex information are based on the 119 unique patients.

**Table 3 jpm-11-00735-t003:** Patient characteristics and composition of datasets used in validation for marker swap optimization. All samples were profiled on the NanoString platform. ICU = intensive care unit; ED = emergency department.

Study	Description	Ethical Committee Approval	N	Median Age (IQR)	Female (%)	Country	Bacteria (%)	Virus (%)	Noninfected (%)
INF-03	Patients with viral infection; collected by nasal swab	Ethics Committee Jehangir Clinical Development Centre Pvt. Ltd., Nov 1, 2019	27	28 (24–37)	10 (37)	India	0 (0)	27 (100)	0 (0)
INF-IIS-03	Adult ED patients with suspected sepsis or acute infection	IRB and Ethical Committee of “ATTIKON” University Hospital, Athens, Greece, 163/05-06-08	76	61 (37–77)	45 (59)	Greece	36 (47)	39 (51)	1 (1)
INF-IIS-10	Bacterial-infected patients from ICU	Community Medical Center, Toms River, NJ, IRB # 17-004	42	69 (60–80)	24 (57)	USA	42 (100)	0 (0)	0 (0)
INF-IIS-11	Adult ED patients with suspected sepsis or acute infection	Ethical Board approval Charite Universitaetsmedizin Berlin EA4/167/18	191	72 (57–81)	83 (43)	Germany	151 (79)	32 (17)	8 (4)
INF-IIS-19	Patients with septic arthritis infection	IRB 6 Stanford University, #4947	20	59 (42–66)	6 (30)	USA	11 (55)	0 (0)	9 (45)
INF-IIS-21	Outpatient viral infections	Comite Etico de Investigacion con Medicamentos, Instituto de Investigacion Biomedica de Salamanca (IBSAL) (code PI 2018 11 138)	21	81 (75–86)	8 (38)	Spain	0 (0)	21 (100)	0 (0)

**Table 4 jpm-11-00735-t004:** Clinical diagnostic performance metrics for the 29 original markers (A) and the 29 final markers (B). The classifier used for original 29 markers was Support Vector Machine (SVM) with non-linear (RBF) kernel. The classifier used for new 29 markers was an assemble of multi-layer perceptron neural networks. The classifiers were selected on the basis of the best trade-off between mAUROC in training data (cross-validation) and validation.

	A: Original 29 Markers		B: New 29 Markers	
Metric	Training (Cross-Validation)	Validation	Training (Cross-Validation)	Validation
Bacterial AUROC	0.817	0.926	0.903	0.925
Bacterial LR-	0.059	0.000	0.075	0.044
Bacterial fraction 1 (%)	7.9	2.1	18.2	14.9
Bacterial band 1 sensitivity (%)	99.3	100	98.1	98.3
Bacterial LR+	7.5	Inf	7.5	14
Bacterial fraction 4 (%)	15.6	22.8	24.3	40.8
Bacterial band 4 specificity (%)	95.0	100	92.2	95.6
Viral AUROC	0.856	0.927	0.913	0.921
Viral LR-	0.075	0.05	0.074	0.071
Viral fraction 1 (%)	23.0	35.3	25.0	33.4
Viral band 1 sensitivity (%)	97.5	97.5	97.3	96.6
Viral LR+	10.0	14.8	10	16
Viral fraction 4 (%)	26.5	24.9	28.6	22.5
Viral band 4 specificity (%)	93.4	95.3	92.8	96.1

**Table 5 jpm-11-00735-t005:** Key information of the 29 mRNAs used in InSep test. Gene symbol, full names, aliases, entrez gene ID, location, and type are given for each of the 29 mRNAs, together with the directionality and relative change in three comparison pairs: bacterial infection vs. uninfected control (BI vs. UC), viral infection vs. uninfected control (VI vs. UC), and bacterial infection vs. viral infection (BI vs. VI).

Gene Symbol	Full Name	Aliases	Entrez Gene ID	Location	Type	BI vs. UC	VI vs. UC	BI vs. VI
**ARG1**	Arginase 1		383	Cytoplasm	Enzyme	↑↑	↑	↑
**BATF**	Basic leucine zipper ATF-like transcription factor		10538	Nucleus	Transcription regulator	↑↑	↑	↑
**C3AR1**	Complement C3a receptor 1		719	Plasma membrane	G-protein coupled receptor	↑↑	↑	↑
**CD163**	CD163 molecule		9332	Plasma membrane	Transmembrane receptor	↑↑	↑	↑
**CEACAM1**	CEA cell adhesion molecular 1		634	Plasma membrane	Transporter	↑↑	↑	↑
**CTSB**	Cathepsin B		1508	Cytoplasm	Peptidase	↑↑	↑	↑
**CTSL**	Cathepsin L	CTSL1	1514	Cytoplasm	Peptidase	↑↑	↑	↑
**DEFA4**	Defensin alpha 4	HP4, HNP4	1669	Extracellular space	Other	↑↑	↑	↑
**FAM214A**	Family with sequence similarity 214 member A	KIAA1370	56204	Other	Other	↓↓	↓	↓
**FURIN**	Furin, paired basic amino acid cleaving enzyme		5045	Cytoplasm	Peptidase	↑↑	↑	↑
**GADD45A**	Growth arrest and DNA damage inducible alpha	DDIT1	1647	Nucleus	Other	↑↑	↑	↑
**GNA15**	G protein subunit alpha 15		2769	Plasma membrane	Enzyme	↑↑	↑	↑
**HK3**	Hexokinase 3		3101	Cytoplasm	Kinase	↑↑	↑	↑
**HLA-DMB**	Major histocompatibility complex, class II, DM beta	RING7	3109	Plasma membrane	Transmembrane receptor	↓↓	↓	↓
**IFI27**	Interferon alpha inducible protein 27		3429	Cytoplasm	Other	↑	↑↑	↓
**ISG15**	ISG15 ubiquitin like modifier		9636	Extracellular space	Other	↓	↑	↓
**JUP**	Junction plakoglobin		3728	Plasma membrane	Other	↓	↑	↓
**KCNJ2**	Potassium inwardly rectifying channel subfamily J member 2		3759	Plasma membrane	Ion channel	↑↑	↑	↑
**LY86**	Lymphocyte antigen 86		9450	Plasma membrane	Other	↓	↑	↓
**NMRK1**	Nicotinamide riboside kinase 1	NRK1, O9orf95	54981	Cytoplasm	Kinase	↑↑	↑	↑
**OASL**	2′-5′-oligoadenylate synthetase like		8638	Cytoplasm	Enzyme	↑	↑↑	↓
**OLFM4**	Olfactomedin 4	GW112, HGC1, OLM4	10562	Extracellular space	Other	↑↑	↑	↑
**PDE4B**	Phosphodiesterase 4B		5142	Cytoplasm	Enzyme	↓↓	↓	↓
**PER1**	Period circadian regulator 1	HPER1, KIAA0482	5187	Nueleus	Transcription regulator	↑↑	↓	↑
**PSMB9**	Proteasome 20S subunit beta 9	RING12, LMP2, PRAAS3	5698	Cytoplasm	Peptidase	↓	↑	↓
**RAPGEF1**	Rap guanine nucleotide exchange factor 1	GRF2, C3G	2889	Cytoplasm	Other	↓↓	↓	↓
**S100A12**	S100 calcium binding protein A12	CAAF1	6283	Cytoplasm	Other	↑↑	↑	↑
**TGFBI**	Transforming growth factor beta induced	CDGG1, CDB1, RDD-CAP	7045	Extracellular space	Other	↓↓	↓	↓
**ZDHHC19**	Zinc finger DHHC-type palmitoyltrasferase 19	DHHC19	131540	Cytoplasm	Enzyme	↑↑	↑	↑

## Data Availability

Datasets used in Table 2 are in data repository GEO (https://www.ncbi.nlm.nih.gov/geo/, accessed on 22 March 2021) or ArrayExpress (https://www.ebi.ac.uk/arrayexpress/, accessed on 22 March 2021). Column “Study” of Table 2 lists accession number for each dataset.

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
