# Peer review of "The Optimization and Biological Significance of a 29-Host-Immune-mRNA Panel for the Diagnosis of Acute Infections and Sepsis"

_jpm, 2021, doi:10.3390/jpm11080735_

Round 1

Reviewer 1 Report

In the presented manuscript, the authors investigate the utility of an updated mRNA panel in the diagnosis of acute infections and sepsis. The article is well written and clearly presents the results of high clinical relevance.

However, the authors should address the following issues:

  1. Can you provide demographic data of the patients included and results of basic diagnostic lab tests for sepsis? Please also provide a comparison of diagnosis of infection using conventional lab tests vs the presented method.
  2. Twenty-nine diagnostic markers seem to be a great number of markers. In this study, you started with a new screening procedure aiming to identify 29 out of 64 genes. Why did you need to limit the number 29? Please explain why you did not consierder a smaller number of genes, or compare several panels with different number of genes?
  3. In the first paragraph in the introduction, causal pathogens mentioned are virus, fungus, or parasites in addition to bacteria. However, there are only 7 genes in your panel distinguishing bacterial from viral infection. How did you detect parasites or fungal infection? Is the assay provided able to detect such cases too? Please provide results and a critical discussion!
  4. Can you provide the predictive data of 30-day mortality rate (such as Kaplan-Meier curve)?
  5. Reference number 13 is not cited correctly

Author Response

Please see the attached document jpm-1265325-response for our point-by-point response. Thank you.

Reviewer 2 Report

He et al. describe procedures for diagnostic and prognostic testing in sepsis using a 29-mRNA qRT-LAMP based assay. 

The overall aim and hypothesis of the study is not yet sufficiently developed - see e.g. Abstract "Altogether, our analysis provides a coherent picture for how analytical performance and data-driven insights can combine to make biologically and clinically relevant diagnostic signatures."

Further, I was not able to retrieve the information on the cohort and samples (material, ethics, groups, etc.) and the chemistry (wet-lab). 

Author Response

(The authors gave the same response as above.)
